# Source apportionment and quantification of liquid and headspace leaks from closed system drug-transfer devices via Selected Ion Flow Tube Mass Spectrometry (SIFT-MS)

**Amos Doepke** *, **Robert P. Streicher**

Health Effects Laboratory Division (HELD), Centers for Disease Control and Prevention (CDC), National Institute for Occupational Safety and Health (NIOSH), Chemical and Biochemical Monitoring Branch (CBMB), Alice Hamilton Laboratories, Cincinnati, Ohio, United States of America

* adoepke@cdc.gov

## Abstract

A system to differentiate and quantify liquid and headspace vapor leaks from closed system drug-transfer devices (CSTDs) is presented. CSTDs are designed to reduce or eliminate hazardous drug (HD) exposure risk when compounding and administering HDs. CSTDs may leak liquid, headspace, or a mixture of the two. The amount of HD contained in liquid and headspace leaks may be substantially different. Use of a test solution containing two VOCs with differences in ratios of VOC concentrations in the headspace and liquid enables source apportionment of leaked material. SIFT-MS was used to detect VOCs from liquid and headspace leaks in the vapor phase. Included in this report is a novel method to determine the origin and magnitude of leaks from CSTDs. A limit of leak detection of 24 μL of headspace vapor and 0.14 μL of test liquid were found using Selected Ion Flow Tube Mass Spectrometry (SIFT-MS).

## Introduction

The National Institute for Occupational Safety and Health (NIOSH) recommends using closed system drug-transfer devices (CSTD)s to limit occupational exposure to hazardous materials and sharps when compounding and administering hazardous drugs (HD)s [1]. CSTDs mechanically prohibit the transfer of environmental contaminants into the system and the escape of hazardous drug (HD) or vapor concentrations outside the system [1]. An engineering challenge associated with CSTD vial adaptor design has been management of the headspace that is either compressed or displaced when transferring liquids in and out of rigid drug vials. CSTD designs and components employ various technologies to manage the displaced headspace volume which can be broadly categorized into two main types, physical barriers type, or air-cleaning type CSTDs [2–4].

USP 800 states that CSTDs must be used for the administration of antineoplastic HDs when the dosage form allows [5]. USP 800 guidelines also recognized the need to evaluate the performance

**Data Availability Statement:** NIOSH Dataset RD-1023-2021-0 Available at Source Apportionment and Quantification of Liquid and Headspace Leaks

from Closed System Transfer Devices via Selected Ion Flow Tube Mass Spectrometry (SIFT-MS) | NIOSH | CDC https://www.cdc.gov/niosh/data/default.html.

**Funding:** The author(s) received no specific funding for this work.

**Competing interests:** The authors have declared that no competing interests exist.

of CSTDs via peer-reviewed studies and demonstrated contamination reduction. NIOSH developed a draft protocol to test material containment of barrier type CSTDs in 2015 [2]. In 2016, NIOSH presented a draft plan to update the testing protocol so that it was applicable to both barrier and air cleaning types of CSTDs [3]. Barrier type CSTDs have been designed to contain air, and it is reasonable to conclude that a headspace leak with a barrier type CSTD would contain the HD at the same concentration as the headspace inside the vial. Air-cleaning type CSTDs allow passage of air to the environment but are tasked with removing hazardous vapors from the exiting air [6].

Previous work to quantify CSTD leaks included a variety of test agents [7], detection methods [8, 9], and testing goals [10–12]. Cyclophosphamide was used as a marker for CSTD contamination reduction when measuring surface contamination in HD preparation areas of a hospital pre- and post-intervention of CSTD use [13]. Jorgenson et al. were among the earlier works that differentiated headspace and liquid leak tests, using titanium tetrachloride as a visible indicator of vapor leaks and in a separate test, sodium fluorescein as an indicator of liquid leaks [14]. Gonzalez et al. continued the use of fluorescein to make measurements of liquid leaks by observing droplets generated during simulations of preparation of HDs while using CSTDs [15, 16]. Queruau et al. used quinine as fluorescent marker [17]. Isopropyl alcohol (IPA) has been used as leak marker; while both liquid and vapor leaks could produce an IPA response, the origin of the leak source, liquid or headspace, was not discernable [2, 18]. Wilkinson et al. used a solution containing propylene glycol methyl ether (PGME) as a leak test agent [19]. PGME was used as a quantifiable marker of liquid leaks, though the ability to detect headspace leaks via PGME was not demonstrated [19]. Hydrogen gas was used as a leak agent as a measure of the gas-tight seal of CSTDs by Besheer et al. [12]. Pressure compatibility by Ishimaru et al. showed that pressure testing could reveal headspace leaks [20, 21].

The CSTD evaluation herein involved operation of CSTDs during a task of transferring a solution between two drug vials as described by Hirst et al. [2]. A test solution containing two volatile organic compounds (VOCs), acetone and methyl tert-butyl ether (MTBE), was used in the evaluation. The VOCs from headspace leaks were vapors, while VOCs from liquid leaks rapidly volatilized. Leaks were measured by detecting vapor phase VOCs in a glove chamber using Selected Ion Flow Tube Mass Spectrometry (SIFT-MS) as the detector. Liquid and headspace leaks were differentiated by the ratios of the two VOCs. The compounds, acetone and MTBE, at equal concentrations in a test solution have a concentration ratio in the headspace vapor of the test solution that is very different, as predicted by their Henry's constants. The ratio of acetone to MTBE detected in the glove chamber can be used to elucidate the source, liquid or headspace. The quantification of the compound (acetone or MTBE) provides the magnitude of a leak. The analytical strategy is similar to stable isotope mixing models used to determine contributions from various sources by measuring isotopic ratios [22].

1,2-propanediol (propylene glycol, PG) was included in the testing solution as a surrogate for a semi-volatile HD component, though it was not quantified. Previously we had proposed two HD surrogates, PG as a marker for liquid leaks and a second HD surrogate (tentatively 1,1,3,3-tetraethylurea (TEU)) as a marker for headspace vapor leaks [3]. In headspace leaks, there was very little PG, while TEU was readily detectable. Both PG and TEU have relatively similar vapor pressures of 0.129 mmHg and 0.208 mmHg, respectively. Slow evaporation from liquid leaks, adsorption to the walls, and consistency of vapor phase PG measurements at ambient conditions made PG difficult to use as an analytical marker of liquid leaks. We felt that extending the sampling time to wait for complete and quantitative evaporation of the PG was not a workable scenario. While it may be possible to quantify with semi-volatile compounds, we opted to differentiate leaks using the rapidly quantifiable dual-VOC, SIFT-MS system presented herein. SIFT-MS offers low limits of detection and real-time response. The real-time response has the benefit of enabling leaks to be temporally correlated with tasks involving

manipulation of CSTD components. Fluorescein was included as a visual qualitative indicator of a liquid leak if present.

## Materials and methods

The test solution, referred to as PGAB solution, contained 2 M 1,2-propanediol (Sigma-Aldrich, St. Louis, MO), 0.33 M acetone (Fisher Co., Fair Lawn, NJ), 0.33 M methyl tert-butyl ether (MTBE) (Baxter Health Care, Muskegon, MI) and 1.3 mM sodium fluorescein (Sigma-Aldrich) in deionized water. Gastight syringes were from the Hamilton Company (Reno, NV). A Cole-Parmer (Vernon Hills, IL) syringe pump was used to deliver the liquid and headspace aliquots. SIFT-MS was done on a Voice 200 Ultra from Syft Technologies (Christchurch, New Zealand) fitted with a HPI hex inlet (with a fixed 20 mL/min inlet flow rate). A Secador® Techni-dome® 360 Large Vacuum Desiccator from Bel-Art Products (Pequannock, NJ) was customized by adding a 30 cm tall cylinder the same diameter as the desiccator between the desiccator halves. Glove ports (20 cm dia.) in the cylinder enabled the desiccator to be used as a glove chamber with a volume of 131 liters. A fan was used to circulate air within the chamber. The SIFT-MS was connected to the chamber via a 4 inch (1/8 inch dia.) length of perfluoroalkoxy (PFA) tube. A figure of the chamber is included in the S1 Fig.

Calibration of the SIFT-MS to vapor phase acetone and MTBE was done by using gas tight syringes to measure 5 µL of neat acetone or MTBE, which was injected into Tedlar bags containing known volumes of air to make vapor phase stocks. Aliquots of the vapor phase stock were then diluted into secondary Tedlar bags of air to achieve the desired concentrations of acetone and MTBE. The reagent-ion reaction rates and the product-ion ratios in the Syft analysis library were adjusted to align the calculated instrument response to the known concentrations of acetone and MTBE [23, 24]. Additional SIFT-MS settings and library information can be found in the S1 Table.

The SIFT-MS response to releases of liquid and headspace aliquots of PGAB solution was done inside the chamber to create calibration curves. Calibration of SIFT-MS response to volumes of liquid leaks was done by releasing liquid aliquots of PGAB solution delivered via gastight syringes and a syringe pump into the chamber where the resulting vapor phase concentrations were measured by SIFT-MS.

Calibration of headspace leaks was done by taking aliquots of headspace, from a 0.5 liter Tedlar bag containing 250 mL of PGAB solution and 250 mL of headspace, into a gas-tight syringe fitted with a closure valve. A 50-mL gastight syringe was used for the upper range of headspace volumes and a 1-mL syringe was used for the range of lower volumes. All experiments were equilibrated at room temperature of 21.5±0.5˚C.

## Results

### Liquid calibration of PGAB solution

Calibration curves were made from liquid aliquots of PGAB solution released inside the chamber, and acetone and MTBE measured by SIFT-MS. For the remainder of the document acetone will be referred to as A, and MTBE referenced as B. The SIFT-MS response to liquid aliquots of PGAB solution is shown in Fig 1. The A and B evaporated from the liquid aliquots of PGAB solution were measured as instrument response reported as vapor concentrations in the chamber.

The standard error of the mean (SEM) for measures of A or B is $s/\sqrt{n}$ where $s$ is the standard deviation and $n$ is the number of measurement points recorded by the instrument. To keep the relative uncertainty of the instrumental measurement as low as possible it is important to include n = 20 to 25 measurements as data points when calculating the average

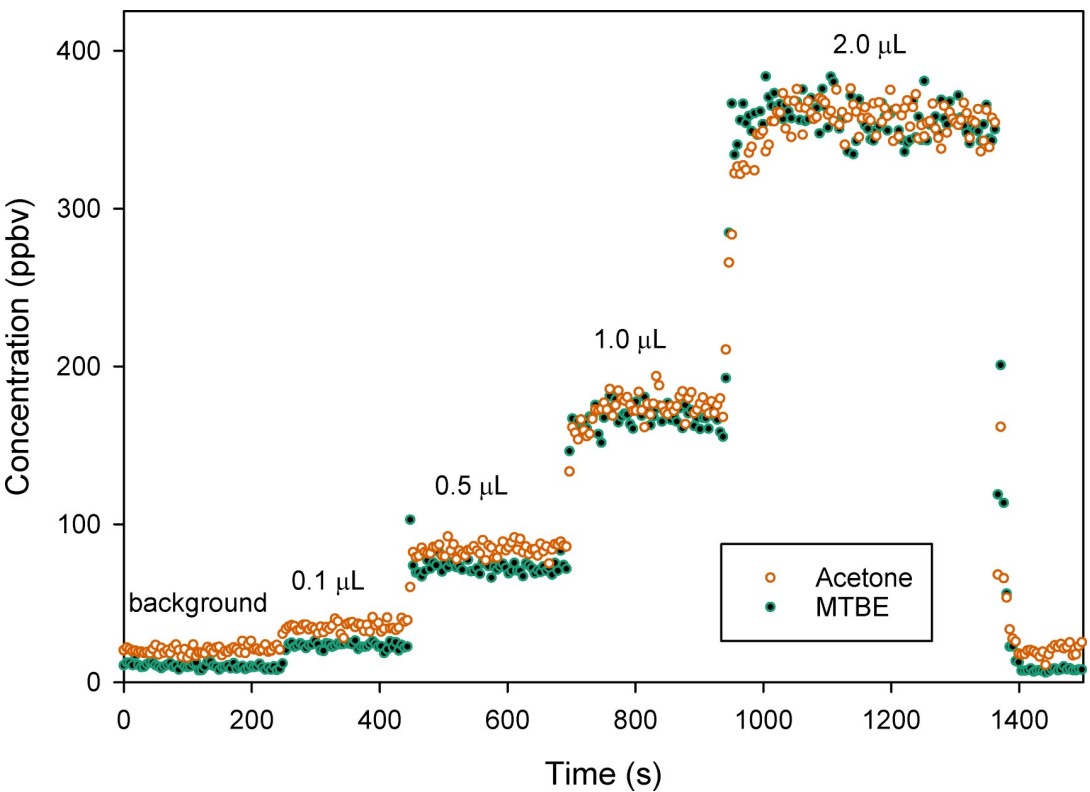

**Fig 1. Chromatogram of SIFT-MS concentration (ppbv) response to acetone and MTBE from aliquots of PGAB solution released in the glove chamber versus time.** The labels indicate background regions and aliquot volumes released.

concentration of A or B from raw data like that obtained in Fig 1. The time required for concentrations of the VOC to stabilize was relatively short. However, the amount of time to collect enough data points ($n$) at a stable concentration to make precise measurements was 200 to 250 seconds after releasing a calibration aliquot.

After baseline subtraction, the instrument response, as a change in the acetone ($\Delta A$) and MTBE ($\Delta B$) concentrations versus the aliquot volume (Fig 2) has a strong linear correlation based on $r^2$ values. The linearity in Fig 2 was typical of individual calibration curves produced in this manner.

The $\Delta A$ and $\Delta B$ were recorded for six (n = 6) independent replicates of 0.5, 1, 10 and 20 μL aliquots of PGAB solution using a 50 μL syringe for an upper calibration range and six independent replicates of 0.1, 0.5, 1 and 2 μL aliquots using a 10 μL syringe as a lower calibration range over multiple days. Daily variation accounts for a significant portion of the method uncertainty when compared to the uncertainties from a single day calibration which is shown in Fig 2. The $\Delta B$ for six multi-day replicates of the upper and lower (inset) calibration ranges are shown in Fig 3, where the dashed line is the linear regression of all 6 replicates and the solid lines are the 95% confidence intervals of the slope. Plot S2 provided in the supplementary information is the liquid calibration curves for $\Delta A$.

## Headspace vapor calibration of PGAB solution

Headspace calibration data was produced by delivering headspace vapor aliquots in six replicates of 0.5, 1, 10, 20 and 50 mL, using a 50 mL syringe, which comprised the upper range

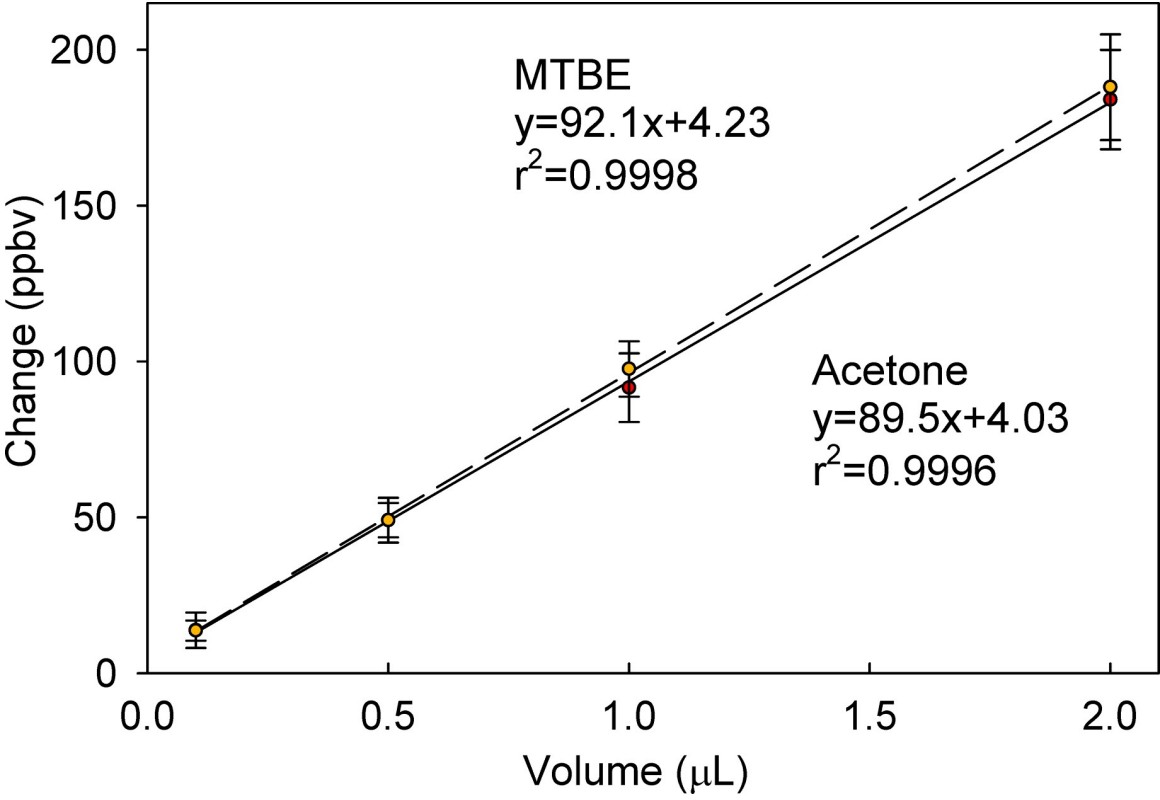

**Fig 2. Calibration curves showing the linear fit of mean instrumental response for MTBE (dashed) and acetone (solid) versus liquid aliquot volumes injected into the chamber.** The uncertainty (error bars) was representative of the standard error of the mean for the measurement region after the instrument response had stabilized from release of a calibration aliquot.

calibration curve. The low range calibration curve was comprised of 0.05, 0.1, 0.2, and 0.5 mL volumes (n = 6) of headspace via a 1 mL syringe. (Headspace calibration curves showing the $\Delta A$ and $\Delta B$ are provided in the S3 and S4 Figs). Calibration data for this method, aggregated from calibration curves produced over multiple days, is shown in Table 1. The slope (m), intercept (b) and their standard deviations (SD), standard error of the slope (SE), $r^2$ and the number of sample replicates (n) used in fitting are listed. The LOD was defined as *3SE/m* and the LOQ as *10SE/m*.

Theoretically, a 0.33 molar solution of a VOC fully vaporized in a 131-liter chamber would produce a chamber concentration of 61 ppbv per µL of solution vaporized. The slopes from the liquid calibration curves were from 71 to 87 ppbv/µL for acetone and MTBE. This was most likely due to differences in tuning the SIFT-MS acetone and MTBE response, which will vary from instrument to instrument and calibration to calibration.

## Ratio of A/B from slopes of the calibration curves

The ratio of $\Delta A$ to $\Delta B$ (A/B) for a liquid ($R_L$) or a headspace ($R_H$) leak can be determined from the ratio of the slopes of the respective liquid and headspace calibration curves. For the lower range calibration plot, the liquid ratio of A/B was 0.92±0.02 (where the error was propagated from the standard deviation (SD) of the slopes) and the headspace ratio of A/B was 0.091 ±0.002. For the upper range calibration plot, the liquid ratio of A/B was 0.92±0.04 (where the error was propagated from the SD of the slopes) and the headspace ratio of A/B was 0.060 ±0.001.

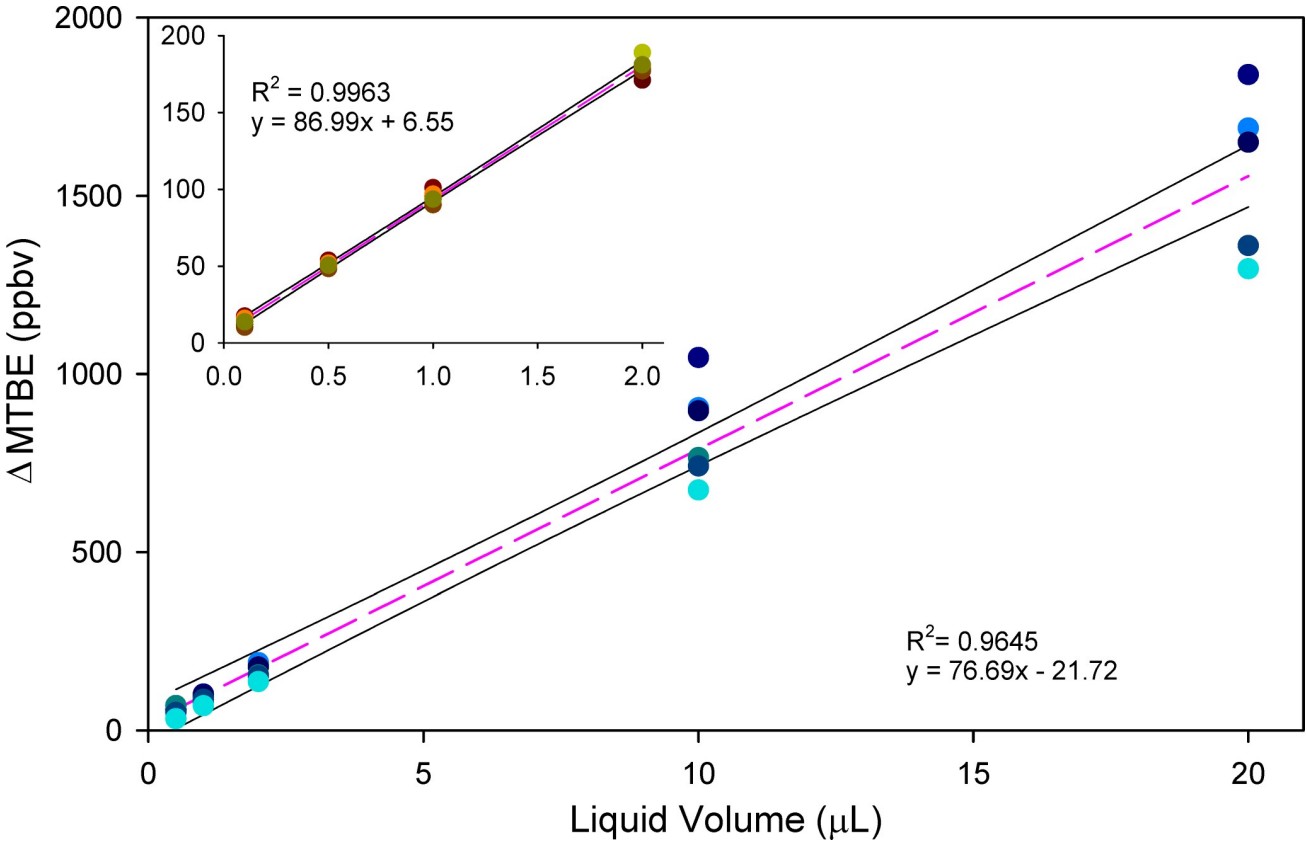

**Fig 3. Linear fitting (dashed) of six replicates of upper range and lower range (inset) calibration data showing ΔB response versus liquid aliquot volume from calibrations.** The aggregated calibration data from replicates of calibration procedures were representative of the variability encountered with the SIFT-MS measurements done over multiple days.

## Average ratio of A/B from all calibration measurements

The ratios of A/B taken from the raw ΔA and ΔB at each aliquot volume can be calculated. The mean ratio (and SD as error bars) at each aliquot volume for liquid and headspace is shown in Fig 4. The mean ± SD of $R_L$ and $R_H$ of all calibration points is shown as solid and dashed horizontal lines. The mean ± SD for the liquid was 0.96±0.11 and 0.094±0.021 for the headspace.

The $R_L$ and $R_H$ agree when calculated as ratios of the slopes or when calculated as an aggregate of raw changes in A and B, though there are large differences in their associated uncertainties. The standard deviation from the aggregated A/B data (Fig 4) were used as limits for determining ratios corresponding to pure liquid or vapor leaks. The ratio of the slopes from

**Table 1. Calibration curve values for the liquid and headspace response curves.**

| Curve | Upper Range Response Curve | | | | | Lower Range Response Curve | | | | | | |
| --- | --- | --- | --- | --- | --- | --- | --- | --- | --- | --- | --- | --- |
| | Slope (m) ± SD | Intercept (b) ± SD | SE | R² | n | Slope (m) ± StDev | Intercept (b) ± SD | SE | LOD | LOQ | R² | n |
| **Headspace Acetone** | 192±1 (response/mL) | -1.6±35 | 144 | 0.9984 | 30 | 180±2 (response/mL) | 1.4±0.5 | 1.50 | 0.0249 (mL) | 0.083 (mL) | 0.9979 | 24 |
| **Headspace MTBE** | 3186±62 (response/mL) | -5242±1531 | 6261 | 0.9893 | | 1981±27 (response/mL) | 15±8 | 23.4 | 0.0354 (mL) | 0.118 (mL) | 0.9958 | |
| **Liquid Acetone** | 71±1 (response/μL) | 25±8 | 33.4 | 0.9964 | 30 | 80.2±1.8 (response/μL) | 6.3±1.8 | 6.35 | 0.238 (μL) | 0.792 (μL) | 0.9887 | 24 |
| **Liquid MTBE** | 77±3 (response/μL) | 22±28 | 114 | 0.9645 | | 87.0±1.1 (response/μL) | 6.6±1.3 | 3.91 | 0.135 (μL) | 0.449 (μL) | 0.9963 | |

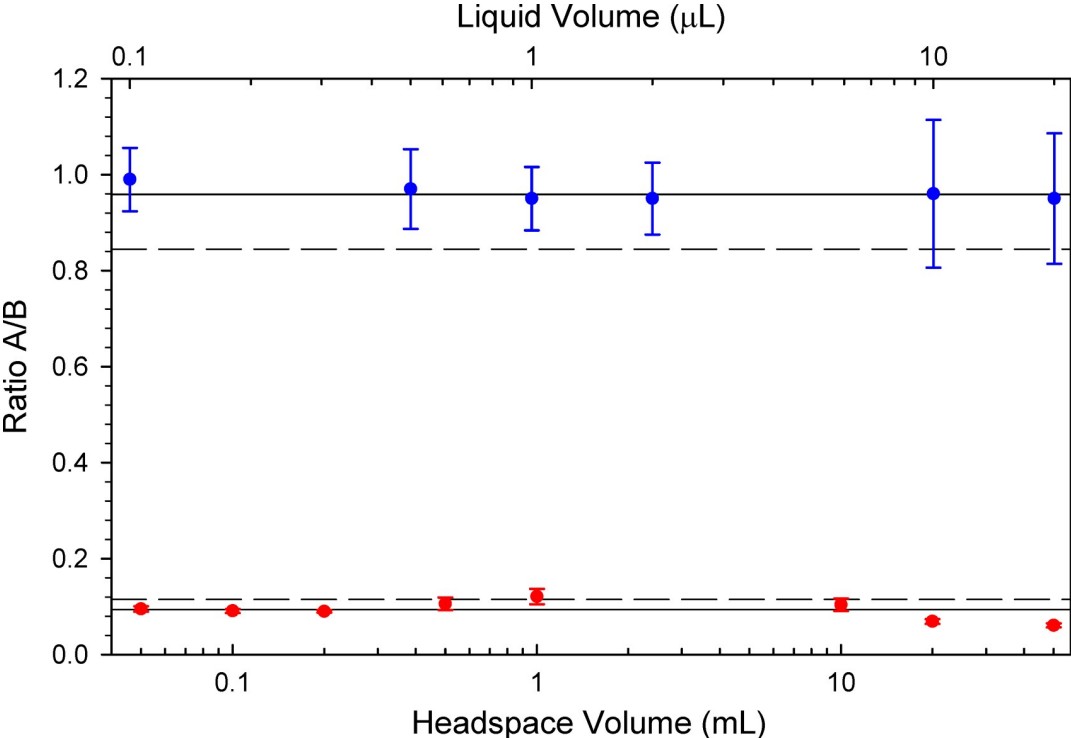

**Fig 4. Mean ratio of A/B versus volume of liquid (upper solid line) and headspace (lower solid line) for aliquots of various volumes (dots).** A mean ratio plus/minus the standard deviation (dashed lines), were considered purely liquid or headspace leaks. While a ratio of A/B (e.g. 0.6) between standard deviations, indicates a leak consisting of a mixture of liquid and headspace.

the calibration curves and their standard deviation should be used in propagation of the error of the leak volumes (Eq 1 through 4).

## Determination of leak type and quantity

The calibration curves can be used to determine the quantity of leak. The calibration ratios were used to determine the liquid or headspace origin of a leak. If the observed ratio of A/B for a leak was within the mean ratio plus/minus a SD of either a pure liquid leak or a pure headspace leak, then the leak could be considered purely liquid or headspace and the volume could be determined directly from the calibration curves.

Leaks that were liquid and headspace mixtures have a ratio of A/B in between that of a pure liquid and pure headspace leak. The following equations are used to proportion the acetone (A) and MTBE (B) masses to liquid and headspace leaks:

$$\Delta B_H = [\Delta A_T - (\Delta B_T \cdot R_L)]/(R_H - R_L) \tag{Eq 1}$$

$$\Delta B_L = [\Delta A_T - (\Delta B_T \cdot R_H)]/(R_L - R_H) \tag{Eq 2}$$

$$\Delta A_H = [(\Delta B_T \cdot R_L) - \Delta A_T]/[(R_L/R_H) - 1] \tag{Eq 3}$$

$$\Delta A_L = [(\Delta B_T \cdot R_H) - \Delta A_T]/[(R_H/R_L) - 1] \tag{Eq 4}$$

where,

$\Delta A_T$ is the change in total detected response of acetone during a time period.

$\Delta A_L$ is the change in response of acetone associated with a liquid leak during a time period.

$\Delta A_H$ is the change in response of acetone associated with a headspace leak during a time period.

$\Delta B_T$ is the change in total response of MTBE during time period.

$\Delta B_L$ is the change in response of MTBE associated with a liquid leak during a time period.

$\Delta B_H$ is the change in response of MTBE associated with a headspace leak during a time period.

$R_L$ is the response ratio of acetone to MTBE in a liquid leak ($m_{AL}/m_{BL}$).

$R_H$ is the response ratio of acetone to MTBE in a headspace leak ($m_{AH}/m_{BH}$).

The values of $\Delta A_T$ and $\Delta B_T$ are taken from the SIFT-MS. The values of $R_L$ and $R_H$ are determined through calibration with standards. This leaves the values of $\Delta A_L$, $\Delta A_H$, $\Delta B_L$, and $\Delta B_H$, 4 unknowns, to be determined by solving Eqs 1–4 simultaneously.

Once the acetone and MTBE responses have been apportioned into their liquid and headspace components ($\Delta A_L$, $\Delta A_H$, $\Delta B_L$, and $\Delta B_H$), the volumes of the liquid and headspace leaks are calculated using the following equations:

$$V_L = \Delta A_L/m_{AL} = \Delta_{BL}/m_{BL} \qquad (Eq\ 5)$$

$$V_H = \Delta A_H/m_{AH} = \Delta B_H/m_{BH} \qquad (Eq\ 6)$$

where,

$V_L$ is the volume of liquid leaked.

$V_H$ is the volume of headspace leaked.

$m_{AL}$ is the slope of the calibration curve of acetone response as a function of liquid leak volume.

$m_{BL}$ is the slope of the calibration curve of MTBE response as a function of liquid leak volume.

$m_{AH}$ is the slope of the calibration curve of acetone response as a function of headspace leak volume.

$m_{BH}$ is the slope of the calibration curve of MTBE response as a function of headspace leak volume.

Note that the liquid leak volume $V_L$ can be determined in Eq 5 from either $\Delta A_L/m_{AL}$ or $\Delta B_L/m_{BL}$ and the headspace leak volume $V_H$ can be determined in Eq 6 from either $\Delta A_H/m_{AH}$ or $\Delta B_H/m_{BH.}$

## Leak volumes as a function of ratio

The ratio of A/B will define a leak as liquid, headspace or a mixture. In Fig 5, we show a theoretical relationship of liquid and headspace volumes when using the equations in the previous section. They were calculated from a relatively small theoretical leak of 100 ppbv $\Delta B_T$ and varying amounts of $\Delta A_T$ to achieve ratios A/B between pure liquid and pure headspace. The resulting leak volumes and standard deviation (error bars) are shown in Fig 5 as a function of the ratio of A/B. The mean (vertical solid lines) and standard deviation (vertical dashed lines) for $R_L$ and $R_H$ from Fig 4 are shown in Fig 5, representing ratios treated either as pure liquid or

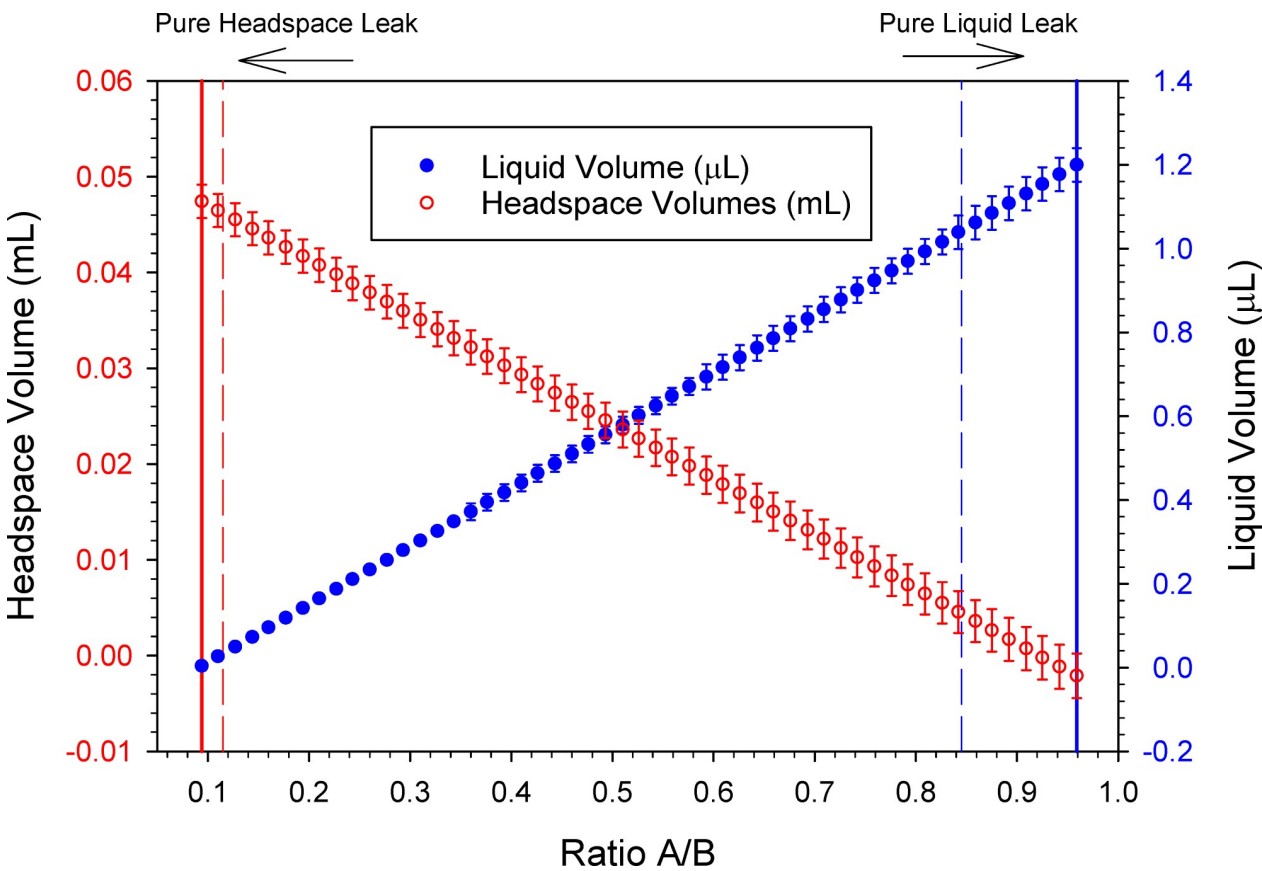

**Fig 5. Theoretical relationship between ratio A/B, the magnitude of the ppbv response, and the volumes of mixed leaks of liquid and headspace derived through equations 1 through 6.** Given 100 ppbv $\Delta B_T$ and varying the amount of $\Delta A_T$ to make ratios across the range of A/B, plotted versus the volume (circles) of headspace and liquid leaks with standard deviation (error bars).

pure headspace leaks. Additional figures describing a theoretical leak with a large, 100 ppmv, $\Delta B_T$ scenarios are shown in S5 and S6 Figs.

### Relative uncertainty as a function of ratio A/B

When solving (Eqs 5 and 6) for volumes of mixed liquid and headspace leaks, the relative propagated error from the calculated volume of headspace ($\sigma V_H$), or volume of liquid ($\sigma V_L$), versus the ratio A/B are shown in Fig 6 for the lower calibration curve data. As the ratio A/B approaches that of a pure liquid, then $\sigma V_L$ approaches 3.4%, while the $\sigma V_H$ increases. Conversely, as the ratio A/B approaches that of a pure headspace leak the $\sigma V_L$ increases, while $\sigma V_H$ approaches 3.76% at the $R_H$. The relative uncertainty in the volume of a leak is large only when that leak represents a small percentage of the total leaked material.

In Fig 6, the $R_H$ at a ratio A/B of 0.0937 is shown as a solid vertical line on the left. The dashed line at 0.115 is $R_H$ plus one standard deviation. The $R_L$ is 0.959 depicted as a solid vertical line and the dashed line is $R_L$ minus a SD at 0.845 (on the right) in Fig 6. The relative percent of the propagated error for $V_L$ and $V_H$ at the significant ratios are shown in Table 2. Leaks with a ratio A/B greater than $R_H+\sigma$ or less than $R_L-\sigma$ should be calculated using the mixed leak equations.

At $R_H+SD$ the 18.6% uncertainty in the volume of a liquid was acceptable being less than 25%. The window of ratios from 0.774 (25% error) to $R_L-SD$ (49.7% error) at 0.845 A/B, where

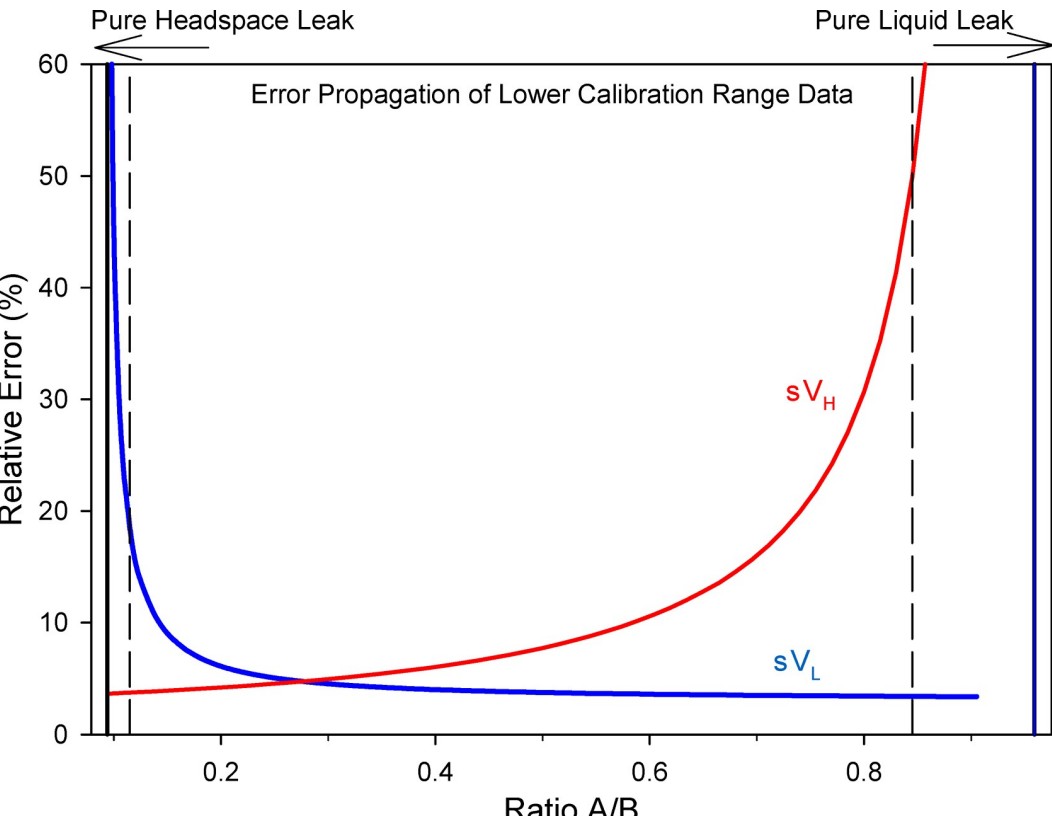

**Fig 6. The relative error versus the ratio A/B for calculated volumes of mixed liquid and headspace leaks.** The standard deviation of instrumental measurements was propagated through the equations used to calculate liquid and headspace volumes from the magnitude and the ratio of the instrumental responses.

the error in $V_H$ is above 25% was relatively broad. However, the volume of headspace leaks in this region, as the ratio approaches $R_L$, is increasingly small.

## Application of method to CSTD testing

A calibration was performed daily to characterize the instrument response before conducting the experiments involving CSTDs. The experimental results presented below involving CSTDs were not meant to be a complete evaluation of any of the devices, but were presented as examples of the data that could be obtained using this method. The test of a CSTD involves operations of the CSTD as would be encountered during normal use procedures. The responses shown in Figs 7 through 9 were obtained while transferring 45 mL of PGAB test solution from one vial to a second vial. The dependent axis is shown in log scale so that changes in the responses are visibly discernable.

**Table 2. Propagated relative percent uncertainty in the calculation of leak volumes at significant ratios of A/B.**

| | Headspace Leaks | | | Liquid Leaks | | |
|---|---|---|---|---|---|---|
| | $R_H$ | $R_H$+SD | R of $\sigma V_L$ = 25% | R of $\sigma V_H$ = 25% | $R_L$-SD | $R_L$ |
| | 0.0937 | 0.115 | 0.108 | 0.774 | .845 | .959 |
| $\sigma V_L$ | 18.6% | 25.0% | 3.45% | 3.41% | $\sigma V_L$ |
| $\sigma V_H$ | 3.76% | 3.71% | 25.0% | 49.7% | $\sigma V_H$ |

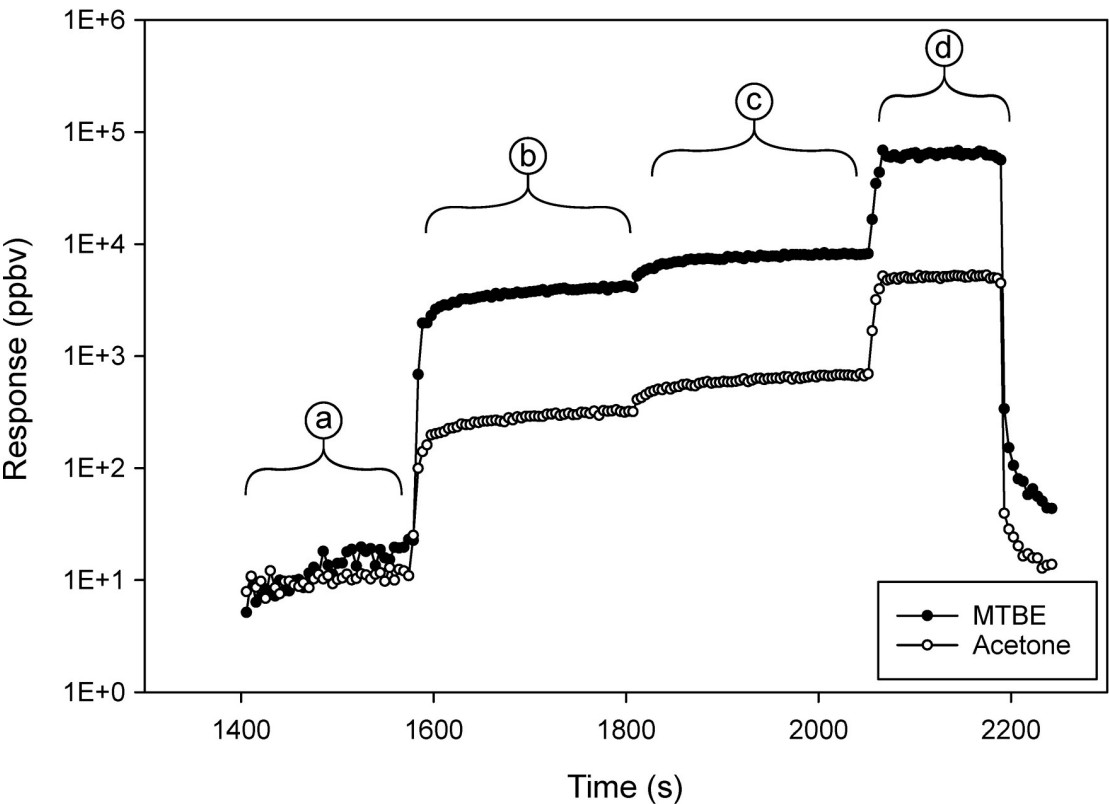

**Fig 7. Instrument response to acetone and MTBE, measured while using an air-cleaning type CSTD to transfer PGAB solution between two vials.** The changes in response in regions (a through d) represent activities at various time points, background collection, attaching vial adaptors, and transferring solution. The decline in signal to near baseline after region (d) was due to opening the chamber post experiment.

In Fig 7, region (a) was background. Region (b) was the response when a CSTD vial-adaptor was placed on the first vial; a headspace release of 3.5 mL vapor was observed. In region (c), a CSTD vial-adaptor was placed on the second vial, with a similar change in response that corresponded to a 3.5 mL release of headspace vapor. At the beginning of region (d), the transfer of 45 mL of PGAB from vial one to vial two occurred. The change in region (d) was calculated to be 43 mL of headspace vapor and no liquid. In total, 50 mL of headspace was observed (Fig 7) when using the air-cleaning CSTD to perform the transfer task. Fluorescein was not visually observed anywhere outside of the CSTD, which supported the information from the A/B measurement indicating the absence of a liquid leak.

In Fig 8, region (a) was background. In region (b), vial-adaptors were placed onto vials one and two. The transfer of 45 mL from one vial to a second vial occurred in region (c). The CSTD continued to release material (d) as the internal pressure equilibrated to the atmosphere. The decrease in concentrations after region (d) was due to evacuation of the chamber using house vacuum. The total response (Fig 8) had a ratio of A/B that was 0.0974, which indicated a headspace vapor leak. The volume was determined to be 3.2 mL of headspace released during the transfer task while using the barrier type CSTD.

A second, barrier type CSTD that employed a bladder to equalize pressure was tested (Fig 9). Region (a) in Fig 9 was background. Region (b) was the response of 0.5 mL headspace from attachment of vial adaptor one. In region (c), the second vial adaptor was placed, resulting in 4.3 mL of headspace vapor. In region (d) the response was measured after

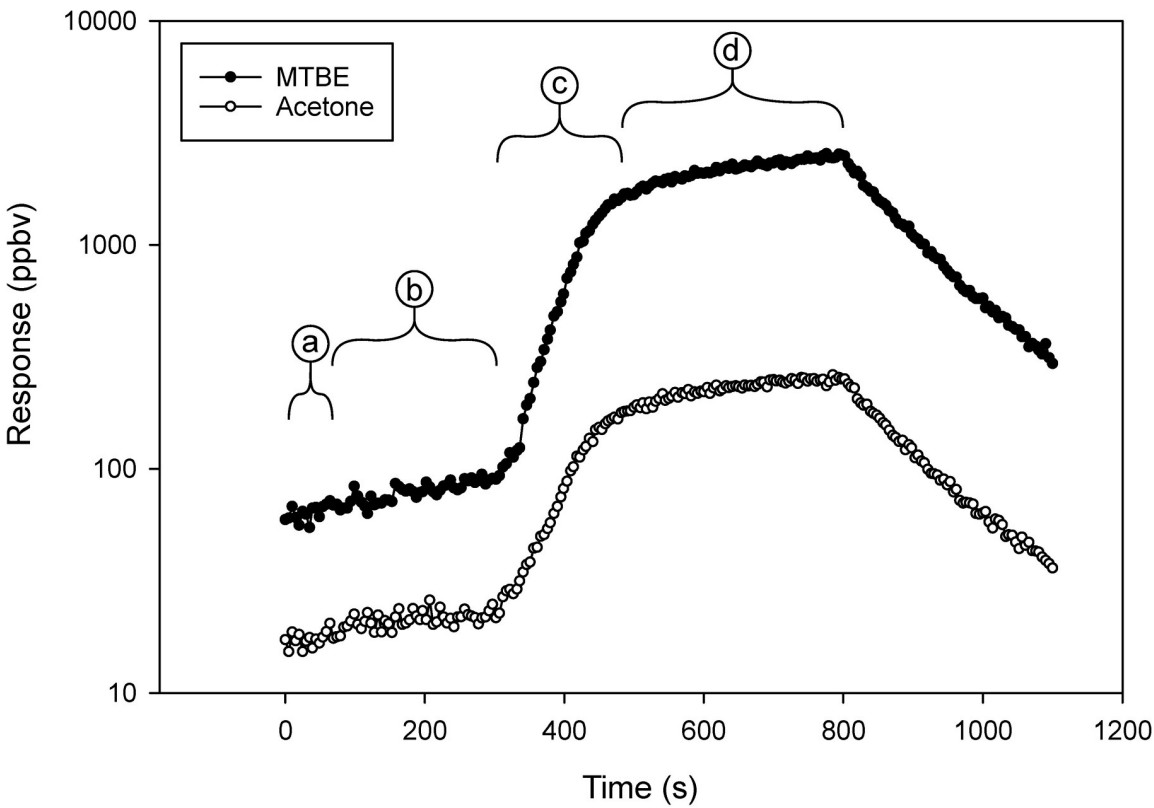

**Fig 8. Instrument response to acetone and MTBE, measured while using a barrier type CSTD to transfer PGAB solution between two vials.** The regions (a through d) represent various time points, which were correlated with activities during the solution transfer process. The decline in signal after region (d) was due to opening the chamber post experiment.

injecting 45 mL of air to expand the equilibration bladder, resulting in 40 mL of headspace vapor released. In the previous step (region (d)), most of the air intended to inflate the bladder leaked out of the CSTD. A new syringe was connected to the CSTD, and in region (e), we tried to inflate the bladder a second time with the same result of the air leaking from the CSTD instead of inflating the bladder. Also, in region (e), 45 mL of PGAB solution was withdrawn from one vial and transferred to a second vial. A release of 166 μL of liquid and 97 mL of headspace vapor was measured. (Note: the CSTD used in the experiments in Fig 9 has been discontinued by the manufacturer.)

## Discussion

Calibration of the SIFT-MS in preparation for assessment of leaks in CSTDs involved calibration of both liquid and headspace leaks for both acetone and MTBE. During calibration of liquid leaks, both the acetone and MTBE response to liquid aliquot volumes were linear across the lower and upper calibration ranges. For calibration of headspace leaks, the acetone response in the headspace was relatively linear across the lower and upper calibration ranges. The headspace MTBE response was linear versus concentration in the lower calibration range. However, the fit for MTBE was not linear at the upper end of the calibration range. This caused some skewing of the A/B ratio for large headspace leaks. To address this non-linearity, we converted the instrument response into mass. For MTBE, a quadratic fit of the log of the response versus the log of the mass was a satisfactory fitting of the relationship between the instrument

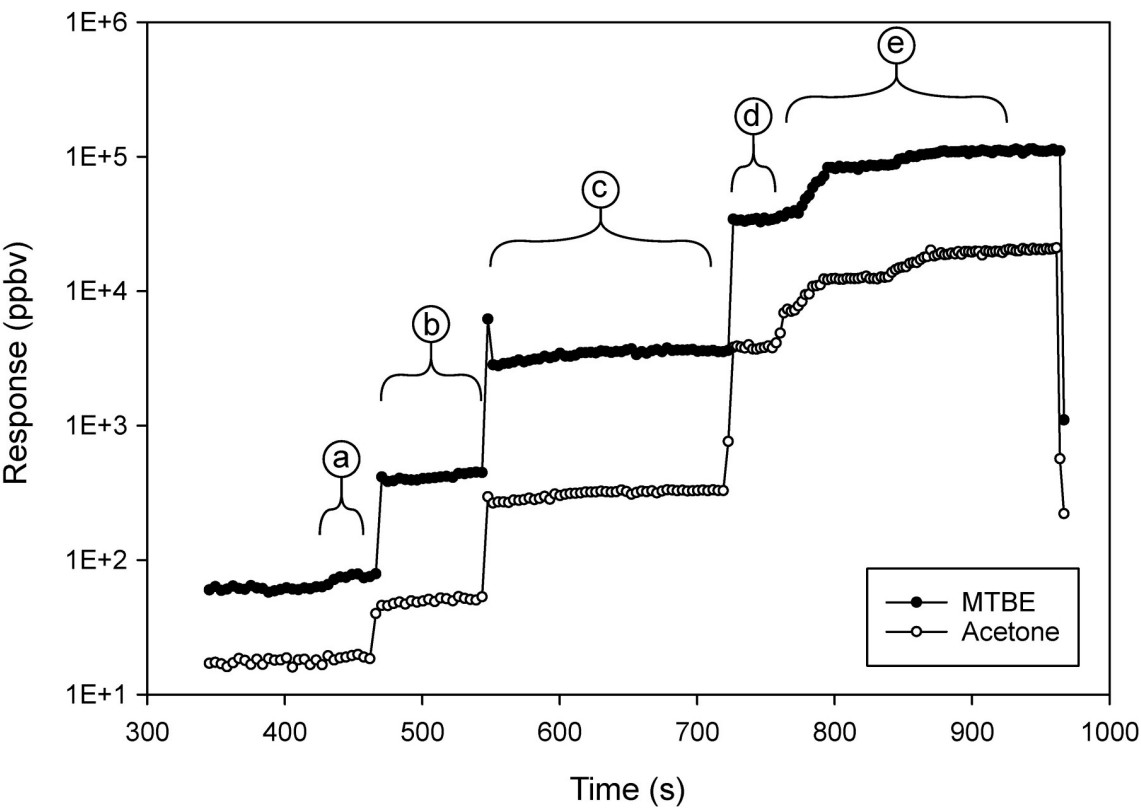

**Fig 9. Instrument response to acetone and MTBE, measured while using a second, barrier type, CSTD to transfer PGAB solution between two vials.** The regions (a through e) represent various time points, which were correlated with activities during the solution transfer process.

response and the mass in the sample. The ratios of the mass of A/B are shown in the S7 Fig, unlike Fig 4, where the ratio was the instrument response of A/B. To mitigate this non-linearity problem in the future, we propose work be done with a molar ratio of A/B at 10 rather than 1 in the liquid by decreasing the MTBE concentration. This will decrease the MTBE in the headspace to a concentration where the instrument response is linear across the range of useful leak volumes. Alternatively, acquiring an adjustable inlet for the SIFT-MS could allow for the reduction of sample volume that enters the SIFT-MS, and subsequently the amount of MTBE would remain in a more linear range of the instrument for the largest leaks.

The quantity of HD in liquid leaks is equal to the volume of the leak, multiplied by the concentration of HD in solution. In contrast, the amount of HD in the headspace over an aqueous HD solution is dependent on the concentration in the solution and the Henry's constant of the HD. HDs would be classified almost exclusively as either non-volatile or semi-volatile. For air-cleaning type CSTDs, the identification of a leak of headspace via detection of VOCs (such as in Fig 7) does not necessarily indicate that the CSTD is ineffective in preventing the escape of a semi-volatile HD that may be more efficiently captured by the air-cleaning device than are VOCs. It only indicates that a volume of headspace has been released. A breakthrough test where air-cleaning devices are tested for the ability to retain compounds with physical properties (e.g. vapor pressure and Henry's constant) similar to HDs is recommended to complement this method. Additionally, a modification to the method described herein would be an important part of the evaluation of air-cleaning type CSTDs. This modification involves isolation of the air-cleaning effluent (headspace VOCs) so that leaks from CSTD connections and other

points of potential failure can be measured with greater precision and sensitivity in the absence of the potentially high VOC background that passes through the air-cleaning CSTD.

We have evaluated the uncertainties associated with quantifying and differentiating liquid and headspace leaks. When mixtures of liquid and vapor leaks occur, the relative uncertainty of the calculated headspace volume increases, and the headspace volume decreases as the ratio A/B approaches a pure liquid leak. Conversely, as the ratio A/B approaches a pure headspace leak, the volume of a liquid leak decreases while the relative uncertainty of the calculation of liquid volumes increases.

The advantage of using a VOC marker system is that the leaks can be detected and quantified in near real-time, allowing temporal correlation between a response and a manipulation of the CSTD. VOCs from liquid leaks evaporate rapidly in the testing chamber, which enables measurement of the VOCs in the vapor phase. For air-cleaning CSTDs, the methods described herein should be complemented with an air-cleaning CSTD breakthrough test designed to challenge the air-cleaning CSTD with the vapor of a surrogate or actual HD. The tests that were shown in this paper were meant to be typical of data obtained using this method. The tests were not intended to be full evaluations of any CSTDs. Presentation of several replicates of CSTD testing would have been required to provide a robust assessment of the performance of CSTDs.

Even the most volatile HDs have low volatility and the concentrations of HDs that exist in the headspace of the solution are many orders of magnitude lower than the concentrations in solution. Based on the estimated Henry's constant for thiotepa (the drug estimated to be the most volatile on the NIOSH Hazardous Drug List [25]) of 2.8 x $10^{-10}$ atm·m$^3$/mole [26], the headspace vapor concentration of a 10 mg/mL solution is estimated to be 1.14 x $10^{-7}$ mg/mL. In contrast, based on the Henry's constant of our test compound acetone of 3.97 x $10^{-5}$ atm·m$^3$/mole, the headspace concentration of our 19.1 mg/mL test solution is approximately 3.1 x $10^{-2}$ mg/mL. Based on these values, had acetone alone been used for leak detection, a 191 μg leak of acetone could either be the result of a 10 μL liquid leak or a 6.16 mL headspace leak. Theoretically, these values would translate to either 100 μg thiotepa in the 10 μL liquid leak or 0.7 ng thiotepa in the 6.16 mL headspace leak released. This demonstrates the importance of differentiating liquid leaks and headspace leaks when evaluating systems used to transfer solutions of semi-volatile drugs.

## Conclusion

Both barrier type CSTDs and air cleaning type CSTDs may be susceptible to either liquid or headspace leaks. The difference in the amount of HD contained in liquid versus headspace vapor leaks may be several orders of magnitude, which is why differentiating a liquid and headspace leak is very important. The work herein is a test method that can detect, differentiate, and quantify headspace and liquid origins of leaks from CSTDs. The real-time nature of the SIFT-MS measurement method enables leaks to be associated with specific tasks and manipulations of the CSTDs. From the detailed analysis of aggregated calibration data, we characterized the instrumental uncertainties of the SIFT-MS. The aggregated calibration data from replicates of calibration procedures and the variability encountered was evaluated. Relative uncertainties of liquid and headspace volume measurements in mixed liquid and headspace leaks are low except for the headspace contribution in nearly pure liquid leaks or the liquid contribution in nearly pure vapor leaks. The SIFT-MS has low limits of detection (pptv-ppbv) for VOCs and semi-volatiles. However, an issue we encountered was inconsistent transfer of leaked semi-volatiles to the instrument for measurement. The semi-volatile surrogates with similar volatility to the most volatile HDs appeared to be subject to adsorption losses on

the surfaces of the experimental chamber rather than being quantitatively transferred to the SIFT-MS. This precluded us from achieving our original goal of measuring a compound with similar volatility to the actual HDs. However, the use of volatile analytes in the test solution enabled near real-time, sensitive measurement without significant adsorption losses, as well as the ability to differentiate liquid and headspace leaks. The composition of a headspace leak from a barrier type CSTD is expected to closely reflect the composition of the headspace within the CSTD. As a result, the procedure described in this paper can adequately assess the efficacy of barrier type CSTDs based on the volume of liquid and headspace vapor leak measured. However, the volatile compounds used in this procedure will readily pass through an air-cleaning CSTD, regardless of its ability to retain a semi-volatile HD. As a result, this method alone cannot be used to assess the effectiveness of an air-cleaning type CSTD to retain a semi-volatile HD that originates in a headspace leak. Assessment of air-cleaning CSTD breakthrough with an appropriate surrogate or actual HD would be required for evaluation of the efficacy of air-cleaning technology CSTDs to remove HD vapors from headspace. After demonstration that an air-cleaning type CSTD adequately contains HD vapor, or that the HD of interest is not sufficiently volatile to present a headspace vapor leak concern, the procedure described herein could be used to assess liquid leaks. Given the high VOC background from passage of VOC vapors though an air-cleaning CSTD, isolation of the effluent from an air-cleaning CSTD filter could improve sensitivity for measurement of liquid leaks and enable measurement of headspace leaks originating from CSTD connections and other points of potential failure in the system. This research provides a framework for future research into assessing the efficacy of CSTDs. In addition to the ability to discriminate between liquid and vapor leaks, future development of these techniques may enable the assessment of containment of HD surrogate vapor by air filtering CSTDs.

## Supporting information

**S1 Fig. Depiction of the chamber, constructed from the desiccator, with the location of the glove ports, circulation fan, work shelf and SIFT-MS sampling port.** The chamber was constructed from a desiccator with custom constructed extender section that included glove ports. A battery powered fan was used to circulate the air.
(PDF)

**S2 Fig. Calibration curves of the liquid PGAB solution.** Calibration data plot of the change in acetone vapor response versus liquid volume aliquot of PGAB solution. Calibration curves were made by releasing aliquots of liquid from PGAB solution in increasing volumes, sampled from the test chamber air.
(PDF)

**S3 Fig. Calibration curves of the headspace of PGAB solution.** Calibration data plot of the change in MTBE response versus headspace aliquot volume from above PGAB solution. Calibration curves were made by releasing aliquots of headspace vapor or liquid from PGAB solution in increasing volumes, sampled from the test chamber air.
(PDF)

**S4 Fig. Calibration curves of the headspace of PGAB solution.** Calibration data plot of the change in acetone response versus headspace aliquot volume from above PGAB solution. Calibration curves were made by releasing aliquots of headspace vapor or liquid from PGAB solution in increasing volumes, sampled from the test chamber air.
(PDF)

**S5 Fig. Plot of volume with error bars representing one standard deviation for a large leak with $\Delta B_T$ at 100 ppmv and $\Delta A_T$ over a range from 90.5 ppmv to 9.5 ppbv to achieve the ratio of A/B over a range from pure headspace leaks to pure liquid leaks.** The resulting leak volume of liquid (VL) and headspace (VH) for a given ratio of A/B are shown on the vertical axes.
(PDF)

**S6 Fig. Plot of volume with error bars representing one standard deviation for a large leak with ΔBT at 100 ppmv and ΔAT over a range from 90.5 ppmv to 9.5 ppbv to achieve the ratio of A/B over a range from pure headspace leaks to pure liquid leaks.** This is the same data as in S4 Fig, but the y axis on the right (liquid leak volume) has been limited to a maximum of 100 μL. The resulting leak volume of liquid leak (VL) and headspace leak (VH) for a given ratio of A/B are shown on the vertical axes.
(PDF)

**S7 Fig. Ratio of mass A / mass B versus volume of liquid (upper line) and headspace (lower line) aliquots.**
(PDF)

**S1 Table. Analysis method settings and configuration of the SIFT-MS.**
(PDF)

## Acknowledgments

The contributions of the following individuals are appreciated; Kenneth K. Brown, Christopher C. Coffey, Crystal D. Forester, Sara Garner, Samuel E. Glover, Lee A. Greenawald, Deborah V.L. Hirst, Kenneth R. Mead, Gabriel A. Merk, Dylan T. Neu, Aaron J. Reeder, Peter B. Shaw, Angela L. Stastny and Emily G. Westbrook.

**Disclaimer:** The findings and conclusions in this report are those of the authors and do not necessarily represent the official position of the National Institute for Occupational Safety and Health, Centers for Disease Control and Prevention. Mention of any company or product does not constitute endorsement by the National Institute for Occupational Safety and Health, Centers for Disease Control and Prevention.

## Author Contributions

**Conceptualization:** Amos Doepke, Robert P. Streicher.

**Data curation:** Amos Doepke.

**Formal analysis:** Amos Doepke.

**Investigation:** Robert P. Streicher.

**Methodology:** Amos Doepke, Robert P. Streicher.

**Supervision:** Robert P. Streicher.

**Writing – original draft:** Amos Doepke.

**Writing – review & editing:** Amos Doepke, Robert P. Streicher.

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
