## [Decision Letter · Decision Letter 0]

27 Jul 2021

PONE-D-21-12702

Source Apportionment and Quantification of Liquid and Headspace Leaks from Closed System Drug-Transfer Devices via Selected Ion Flow Tube Mass Spectrometry (SIFT-MS)

PLOS ONE

Dear Dr. Doepke,

Thank you for submitting your manuscript to PLOS ONE. After careful consideration, we feel that it has merit but does not fully meet PLOS ONE’s publication criteria as it currently stands. Therefore, we invite you to submit a revised version of the manuscript that addresses the points raised during the review process.

We look forward to receiving your revised manuscript.

Kind regards,

Suman S. Thakur, Ph.D

Academic Editor

PLOS ONE

Journal Requirements:

2. Please upload a copy of Supporting Information Figure S8, which you refer to in your text on page 12.

Reviewers' comments:

Reviewer's Responses to Questions

**Comments to the Author**

1. Is the manuscript technically sound, and do the data support the conclusions?

Reviewer #1: Yes

Reviewer #2: Yes

Reviewer #3: Yes

2. Has the statistical analysis been performed appropriately and rigorously? 

Reviewer #1: Yes

Reviewer #2: Yes

Reviewer #3: Yes

3. Have the authors made all data underlying the findings in their manuscript fully available?

Reviewer #1: Yes

Reviewer #2: Yes

Reviewer #3: Yes

4. Is the manuscript presented in an intelligible fashion and written in standard English?

Reviewer #1: Yes

Reviewer #2: No

Reviewer #3: No

5. Review Comments to the Author

Reviewer #1: This is a very thorough study showing how SIFT-MS can be used for leak testing of Closed System Drug-Transfer Devices. The results are based on well designed and carefully carried out experiments. The discussion including aspects such Henrys law and ratios of the two test compounds and observed non-linearity is logical and complete.

The authors revised the manuscript according to the reviewer comments on the original submission.

The explanation on the last page of the Supplemental Materials is more clear now.

It should be mentioned that for this application, reduced sample flow rate would result in better linearity. There is a comment "Commented [DA(3]: Our instrument has a fixed (nonadjustable) sample inlet flow rate. I will mention this in the manuscript." but it is not really discussed. The actual flow rate used should be stated.

Reviewer #2: This manuscript appears to describe a technical method for leak testing of closed system drug-transfer devices using SIFT-MS. The manuscript is a resubmission (following initial rejection), and the previous reviewers' comments are appended, with the author's comments and thoughts included as track changes.

While some of the reviewers comments seem to have been addressed, as reviewer 3 stated "The research is presented more as an internal laboratory report, than as a scientific article targeting a wide range of multidisciplinary readers.". This does not appear to have been addressed in any meaningful way.

Reviewer #3: The authors were well replied to technical comments and the paper was corrected as suggested by the referees. However some specific remarks are mandatory to enhance the quality of this manuscript.

The paper would be easier to understand if the details of each figure and table are explained and discussed (they should be more explained and detailed). In general, the authors provide a somewhat clear description of what was done.

Experimental results are presented in a vague manner. Some justifications and more information about the calibration and quantification results should be given, otherwise this is a serious problem because there is no way to understand how those results are robust and accurate.

The discussion is incomplete or even missing, does not clearly present the authors' conclusions, and make some broad generalizations that are not based upon the data presented. It should be re-written to provide a much clearer interpretation of results comparing them with previous findings in this filed. Relevant references should be included and discussed. In addition, in the conclusion section, a critical evaluation of the advantages and limitations of the SIFT-MS technique, the future perspectives and the expected outcomes in the field of leak testing of Closed System Drug-Transfer Devices are necessary and strongly recommended.

6. PLOS authors have the option to publish the peer review history of their article (what does this mean?). If published, this will include your full peer review and any attached files.

Reviewer #1: **Yes: **Patrik Španěl

Reviewer #2: No

Reviewer #3: No

---

## [Author Response · Author response to Decision Letter 0]

10 Sep 2021

response to reviewers uploaded in documents section

---

## [Decision Letter · Decision Letter 1]

28 Sep 2021

Source Apportionment and Quantification of Liquid and Headspace Leaks from Closed System Drug-Transfer Devices via Selected Ion Flow Tube Mass Spectrometry (SIFT-MS)

PONE-D-21-12702R1

Dear Dr. Doepke,

We’re pleased to inform you that your manuscript has been judged scientifically suitable for publication and will be formally accepted for publication once it meets all outstanding technical requirements.

Kind regards,

Suman S. Thakur, Ph.D

Academic Editor

PLOS ONE

Additional Editor Comments (optional):

Reviewers' comments:

Reviewer's Responses to Questions

**Comments to the Author**

1. If the authors have adequately addressed your comments raised in a previous round of review and you feel that this manuscript is now acceptable for publication, you may indicate that here to bypass the “Comments to the Author” section, enter your conflict of interest statement in the “Confidential to Editor” section, and submit your "Accept" recommendation.

Reviewer #1: All comments have been addressed

Reviewer #3: All comments have been addressed

2. Is the manuscript technically sound, and do the data support the conclusions?

Reviewer #1: Yes

Reviewer #3: Yes

3. Has the statistical analysis been performed appropriately and rigorously? 

Reviewer #1: Yes

Reviewer #3: Yes

4. Have the authors made all data underlying the findings in their manuscript fully available?

Reviewer #1: Yes

Reviewer #3: Yes

5. Is the manuscript presented in an intelligible fashion and written in standard English?

Reviewer #1: Yes

Reviewer #3: Yes

6. Review Comments to the Author

Reviewer #1: This is a very thorough study showing how SIFT-MS can be used for leak testing of Closed

System Drug-Transfer Devices. The results are based on well designed and carefully carried out

experiments. The discussion including aspects such Henrys law and ratios of the two test compounds

and observed non-linearity is logical and complete.

The authors revised the manuscript according to the reviewer comments.

Reviewer #3: The paper was corrected, and all the comments were addressed. It was improved as a result. It could be accepted.

7. PLOS authors have the option to publish the peer review history of their article (what does this mean?). If published, this will include your full peer review and any attached files.

Reviewer #1: **Yes: **Patrik Španěl

Reviewer #3: No

---

## [Editor Report · Acceptance letter]

25 Oct 2021

PONE-D-21-12702R1 

Source Apportionment and Quantification of Liquid and Headspace Leaks from Closed System Drug-Transfer Devices via Selected Ion Flow Tube Mass Spectrometry (SIFT-MS) 

Dear Dr. Doepke:

I'm pleased to inform you that your manuscript has been deemed suitable for publication in PLOS ONE. Congratulations! Your manuscript is now with our production department. 

Kind regards, 

on behalf of

Dr. Suman S. Thakur 

Academic Editor

PLOS ONE